# Molecular Evolution in a Peptide-Vesicle System

**DOI:** 10.3390/life8020016

**Published:** 2018-05-24

**Authors:** Christian Mayer, Ulrich Schreiber, María J. Dávila, Oliver J. Schmitz, Amela Bronja, Martin Meyer, Julia Klein, Sven W. Meckelmann

**Affiliations:** 1Institute of Physical Chemistry, CENIDE, University of Duisburg-Essen, 45141 Essen, Germany; 2Department of Geology, University of Duisburg-Essen, 45141 Essen, Germany; ulrich.schreiber@uni-due.de (U.S.); maria.davila@uni-due.de (M.J.D.); 3Institute of Applied Analytical Chemistry, University of Duisburg-Essen, 45141 Essen, Germany; oliver.schmitz@uni-due.de (O.J.S.); amela.bronja@uni-due.de (A.B.); martin.meyer.m17@stud.uni-due.de (M.M.); julia.klein@uni-due.de (J.K.); sven.meckelmann@uni-due.de (S.W.M.)

**Keywords:** origin of life, evolution, molecular evolution, prebiotic chemistry, peptides, vesicles

## Abstract

Based on a new model of a possible origin of life, we propose an efficient and stable system undergoing structural reproduction, self-optimization, and molecular evolution. This system is being formed under realistic conditions by the interaction of two cyclic processes, one of which offers vesicles as the structural environment, with the other supplying peptides from a variety of amino acids as versatile building blocks. We demonstrate that structures growing in a combination of both cycles have the potential to support their own existence, to undergo chemical and structural evolution, and to develop unpredicted functional properties. The key mechanism is the mutual stabilization of the peptides by the vesicles and of the vesicles by the peptides together with a constant production and selection of both. The development of the proposed system over time would not only represent one of the principles of life, but could also be a model for the formation of self-evolving structures ultimately leading to the first living cell. The experiment yields clear evidence for a vesicle-induced accumulation of membrane-interacting peptide which could be identified by liquid chromatography combined with high-resolution mass spectroscopy. We found that the selected peptide has an immediate effect on the vesicles, leading to (i) reduced vesicle size, (ii) increased vesicle membrane permeability, and (iii) improved thermal vesicle stability.

## 1. Introduction

It is generally accepted that complex prebiotic structures do not appear accidentally, but instead form in a long-term process facilitated by random variation, selection, and reproduction. This process, representing a very general form of evolution, must have dominated even the very early steps, leading to very primitive precursors of a living cell. It must have been based on the building blocks of prebiotic chemistry which potentially have been formed in many different locations on or near the early Earth [1,2,3,4,5,6,7,8,9].

An important fraction of prebiotic compounds is comprised of amphiphilic substances, the precursors of lipids. Their unique tendency to form various mesostructures, most prominently multilayers and vesicles with double-layer membranes, makes them a natural starting point for cell-like compartments [10,11,12]. They also have the capability to select and accumulate other prebiotic molecules [13,14], especially if those molecules are amphiphilic as well [15,16]. Those amphiphilic mesostructures may occur naturally in a variety of environments, such as shore lines, hot springs [11,17], or even in bulk liquid phases.

A very special environment for the formation of amphiphilic structures includes deep-reaching tectonic fault zones [18]. Recently, we proposed a mechanism of periodic vesicle formation which is expected to occur in fault zones filled by water and CO_2_ [15]. At a depth of approximately −1 km, pressure and temperature conditions induce a local phase transition between supercritical CO_2_ (scCO_2_) and subcritical gaseous CO_2_ (gCO_2_). Various amphiphilic products of hydrothermal chemistry [19] are expected to accumulate at this point due to the solubility drop in CO_2_ and the presence of large transition-induced interfaces.

With additional periodic pressure variations resulting from tidal influences or geyser phenomena, a cyclic process occurs in which the transition scCO_2_ → gCO_2_ induces the formation of water droplets covered by a monolayer of amphiphilic compounds [20]. When migrating through the interface to the aqueous domain (which by itself is covered by a layer of amphiphiles), the droplets turn into vesicles with a bilayer membrane [21]. Being thermodynamically unstable, the vesicles are expected to disintegrate and release their organic contents into the bulk water phase over time. During the transition gCO_2_ → scCO_2_, the organic constituents and the water again become soluble in the CO_2_ phase and the cycle can start again [15]. So, in general, each pressure cycle corresponds to one generation of vesicles, even though individual vesicles may survive for several cycles.

In the same hydrothermal environment, amino acids are expected to occur [22,23,24], which are being formed at higher pressure and temperature conditions and are being spread by convectional flow of the fluid media. Under the given temperature and pressure conditions and in presence of the water/carbon dioxide interface, these amino acids undergo spontaneous condensation reactions and form a series of oligopeptides [16,25]. After a short period of time, the competing processes of condensation and hydrolysis will lead to an equilibrium situation. In this state, the concentrations of longer oligopeptides are very small. In a corresponding laboratory experiment, they decrease by approximately one order of magnitude for each additional amino acid unit [16]. Nevertheless, this process leads to a constant presence of random oligopeptides of variable length.

However, if a given peptide with a specific amino acid sequence is capable of interacting with the bilayer membranes of the vesicles described above, e.g., by being amphiphilic with an amphiphilicity profile resembling the one of the membrane, it will integrate into the bilayer structure. Such an integrated peptide is now protected against hydrolysis and therefore will accumulate over time. Consequently, its concentration can keep on growing and may surpass the original equilibrium concentration by several orders of magnitude [16]. At the same time, primarily hydrophilic peptides are recycled and primarily hydrophobic ones will de eluted by scCO_2_ (Figure 1).

Of course, such an accumulation has consequences for the vesicle structure. If the vesicle is being stabilized by the given peptide, its lifetime will increase, maybe even over several pressure cycles. This given, the period of protection for the corresponding peptide will also increase, giving this peptide a further selection advantage over competing peptides. This mutual effect (peptide stabilizes the vesicle—the vesicle stabilizes the peptide) has the capability to drive an ongoing evolution of a peptide-vesicle system, targeting vesicles with an optimized potential to survive the given pressure-cycling conditions. The resulting vesicle system could be the starting point for the subsequent development of a living cell [26,27,28].

In the following, we want to report on an experiment which is meant to promote such an evolution process. It combines the cyclic formation and destruction of vesicles (not of the chemical components thereof) with the conditions of random peptide formation. It involves selection pressure on the vesicle structure and the observation of the optimization process over time. Finally, a resulting peptide/vesicle system is being studied for its characteristic features regarding vesicle stability and its physical properties. Overall, this experiment may be a rare example of a system which develops from a simple towards a much more complex one.

## 2. Materials and Methods

### 2.1. Amphiphiles

The choice of amphiphiles was driven by two motivations: (i) to focus on simple chemical structures which could easily develop in a hydrothermal system; and (ii) to allow for the formation of vesicles which are stable at high temperatures, at a broad pH range, and in the presence of bivalent cations. For the two latter conditions, T. Namani and D. Deamer have proposed a mixture of long chain amines with long chain fatty acids [29] which are accessible by Fischer-Tropsch chemistry [4]. In order to improve temperature stability, C_18_ chains have been chosen for both components. Accordingly, octadecylamine (GC purity ≥ 99.0%) and octadecanoic acid (GC purity ≥ 94.5%) were purchased from Sigma Aldrich (St. Louis, MO, USA). Both amphiphiles are being used in a 1:1 mass ratio.

### 2.2. Amino Acids

The choice of amino acids again followed the condition of being accessible by hydrothermal chemistry. Initially, only proteinogenic amino acids in their natural L-form were considered for simplicity. With that, the selection was limited to a set of 12 L-amino acids which were experimentally accessible under simulated hydrothermal conditions [22]. This includes the hydrophilic amino acids glycine, serine, threonine, aspartic acid, glutamic acid, and lysine, as well as the less hydrophilic or hydrophobic ones alanine, proline, valine, leucine, isoleucine, and phenylalanine. All amino acids (HPLC or titration purity ≥ 98.0%) were purchased from Sigma Aldrich and were used without further purification.

### 2.3. Pressure Cell and Initial Setup

In order to simulate conditions given in depths between 1 and 7 km, the high pressure cell (50 mL) of a custom-made phase equilibrium apparatus (SITEC-Sieber Engineering AG, Ebmatingen, Switzerland) is used as a reaction container. It allows for pressures of up to 1000 bar with manual fine adjustment, elevated temperatures, and constant stirring. It is filled with 25 mL of water (Millipore Milli-Q water, resistivity 18.2 mΩ∙cm) and 25 mL of carbon dioxide (Air Liquide, GC purity ≥ 99.995%), with the latter being either in the gaseous or in the supercritical state depending on the pressure and temperature conditions. A system of valves and pressure regulations allows one to add or remove samples from both phases without changing the pressure inside the cell. All amphiphiles and amino acids were added to the aqueous phase prior to the start of the experiment. The concentrations of the amphiphiles octadecylamine and octadecanoic acid were adjusted to 0.01 M each, and each of the 12 amino acids was added in 0.067 M concentration in the aqueous solution.

### 2.4. Evolution Experiment

In order to accelerate the peptide formation cycle and to induce selection pressure on the vesicles, the temperature inside the cell is kept at 120 °C during the whole experiment. The pressure is repeatedly switched between 100 bar and 70 bar on a regular time scale (every 30 min). During each pressure cycle, a phase transition from supercritical scCO_2_ to gaseous gCO_2_ and vice versa is induced. In each cycle, the transition towards gCO_2_ is accompanied by the appearance of micrometer-sized droplets which, when in contact with the bulk aqueous phase, form vesicles with membranes in the liquid crystalline phase state (see [15] for micrographs). On the other hand, the transition towards scCO_2_ leads to a depletion of amphiphiles in the aqueous phase and therefore to (at least partial) vesicle disintegration. Cyclic pressure changes are induced by a counterbalance piston directly connected to the cell. The total evolution experiment is run for 160 h and involves 85 pressure cycles. Samples of the aqueous phase (300 µL) are taken at t = 0, t = 16 h, t = 44 h, t = 90 h, and t = 160 h. Due to the presence of CO_2_ at high pressure, the aqueous solution is quite acidic, with a pH of around 3.

### 2.5. PFG-NMR Experiments

The PFG-NMR experiments basically follow a scheme which has been applied in earlier studies on vesicle dispersions [30,31]. All corresponding ^1^H diffusion experiments are run on a 500 MHz DRX spectrometer (Bruker BioSpin GmbH, Rheinstetten, Germany) with a Bruker DIFF30 probe head (1200 G/cm maximum field gradient). All measurements are performed at 298 K in a 5 mm Shigemi NMR sample tube, adding 10% D_2_O for spin locking. As a pulse program, the stimulated echo pulse sequence is combined with two gradient pulses. A total number of 16 scans is used for each measurement. The spacing ∆ between the two gradient pulses is set to 25, 50, and 100 ms. The gradients are adjusted to strengths G between 1.5 G/cm and 750 G/cm, and the gradient pulse duration δ is set to 2.0 ms. All resulting echo intensities are plotted logarithmically against the parameter γ^2^G^2^δ^2^(∆-δ/3), with γ being the gyromagnetic ratio of the hydrogen (^1^H) nucleus. In these so-called Stejskal-Tanner-plots, each slope corresponds to the negative diffusion coefficient of the observed system component.

In order to observe the development of the vesicle properties over time, the experiments are repeated over extended time periods. In order to study the effect of the peptide, all measurements are compared with the results on original vesicles. The thermal stability of the vesicles with and without peptides is assessed by introducing storage intervals at elevated temperature between the measurements (T = 50 °C).

### 2.6. Identification of Accumulated Peptides

Each sample (300 µL, see Section 2.4) was centrifuged in order to separate dispersed solid products from the aqueous solution, basically consisting of vesicles. The separated solid products were quantified and dissolved in 150 µL isopropanol. Prior to analysis, these samples were diluted 1:1 (*v*/*v*) with HPLC-water and analyzed along with blank samples (isopropanol/water; 1:1; *v*/*v*) for background subtraction. Both the sample and the blank were analyzed three times. Global peptide analysis was performed on an Agilent 1290 Infinity liquid-chromatography system (Santa Clara, CA, USA) consisting of a 1290 Infinity binary pump (G4220A) with a Jet Weaver V35 mixer, a 1290 Infinity HiP sampler (G4226A), and a 1290 Infinity Thermostated Column compartment (G1316C) coupled to an IM-qTOF-MS (Agilent 6560). Separation was performed on a C18 reversed phase column (Aeris Peptide XB-C18; 150 × 2.1 mm; 1.7 µm; Phenomenex, Aschaffenburg, Germany) using water and 0.1% formic acid as eluent A and acetonitrile acidified with 0.1% formic acid as eluent B. The gradient started at 5% B for 2 min, was linear to 85% B after 20 min, was linear to 95% B after 25 min, and held at 95% B for 10 min. The column was re-equilibrated at the initial conditions for 10 min. The total run time was 45 min at a flow rate of 0.1 mL/min, and the injection volume amounted to 10 µL. The column was kept at room temperature. LC flow was introduced into the IM-qTOF-MS using a dual Agilent Jet Stream Electrospray Ionization (AJS ESI) source operating in the positive mode. Source parameters were as follows: capillary 5000 V, nozzle voltage 500 V, nebulizer gas 20 psi, sheath gas 12 L/min (both N2), dry gas temperature 200 °C, and sheath gas temperature 325 °C. Spectra were recorded in q-TOF only mode between *m/z* 50-3200.

For molecular feature finding and principal component analysis, the MassHunter Qualitative Analysis (B.07.00) was used. Background ions from blank analysis were removed from the resulting feature list using Mass Profiler Professional (12.6.1) to avoid false positive results. Subsequently, the feature list was searched for possible peptides using an in-house software tool.

MS^n^ analysis of possible peptides was carried out on a Merck-Hitachi D-7000 HPLC system (Kenilworth, NJ, USA) equipped with an L-7100 quaternary pump and an L-7200 autosampler coupled to a Bruker Amazon Speed iontrap MS (Bruker Daltonics, Bremen, Germany). LC conditions were the same as described above, except for the flow rate of the mobile phase, which was changed to 0.15 mL/min. ESI parameters were as follows: capillary 5000 V, end plate offset 500 V, nebulizer gas 20 psi, dry gas 3 L/min (both N_2_), and dry gas temperature 200 °C. The MS^n^ experiment was carried out by selecting the precursor ion of *m/z* 864.5 and scanning the resulting fragment ions from *m/z* 200 to 875 after activation with an amplitude of 1.0.

### 2.7. Peptide Synthesis and Vesicle Reconstruction

The peptide H-Lys-Ser-Pro-Phe-Pro-Phe-Ala-Ala-OH (being the largest of the accumulated species) was synthesized commercially by APeptide Co., Ltd. (Shanghai, China). The total amount of 7 mg was used in portions to reassemble the vesicle structure which has led to its accumulation. In each experiment, 2.6 mg of the peptide was added to 100 µL of a vesicle dispersion formed by a 1:1 mixture of 20 millimolar solutions of octadecanoic acid and octadecylamine at pH = 3 (adjusted with 1 N HCl) after six cyclic pressure cycles in the high pressure cell. Subsequently, the resulting vesicle dispersion was submitted to field gradient NMR experiments in order to study their average size, membrane properties, and thermal stability. With the given concentrations, the final volume of 100 µL contains approximately 0.55 mg of the amphiphile and 2.6 mg of the peptide. If 20% of the peptide is integrated into the vesicle membranes, the mass ratio between the peptide and amphiphile is around 1:1 and the molar ratio around 1:3.5.

## 3. Results

### 3.1. Vesicle Formation

The 1:1 mixture of octadecanoic acid and octadecylamine has unique physical properties which affect its tendency to form vesicles depending on the given pH value of the aqueous solution. While no vesicles are being formed at neutral pH, any deviation will either lead to the deprotonation of the carboxylic groups of the octadecanoic acid or to a protonation of the amine, and hence to head group ionization and vesicle formation in both cases. With the given acidic environment, the protonation of the amine will create charged head groups of the amine which initiates the vesicle formation. Experimental evidence for the resulting vesicles during the pressure cycling has already been gained from optical microscopy and PFG-NMR [15,16]. Some of the vesicles obviously show a multilamellar structure and contain internal vesicles, with both features possibly deriving from the fusion of droplets in the gas phase [15,16]. A micrograph of the vesicles formed in the present series of experiments is presented in Figure 2.

The volume-averaged diameter of the vesicles generally depends on the rate of decompression. In the given case, the majority of the vesicles are smaller than one micrometer, and their structure is therefore not resolved by optical microscopy. However, a small fraction of the vesicles shows diameters above 2 µm. Their membrane morphology can be observed under phase-contrast illumination (arrows in Figure 2). In addition, the vesicle structure formed by bilayers of amphiphilic molecules is supported by PFG-NMR measurements, which show the presence of encapsulated water molecules inside membranes of a relatively low permeability (Figure 3).

In this set of plots, the initial, very steep decay marks the free water molecules, and the negative slope corresponds to the self-diffusion coefficient of bulk water. The shallow part of each plot refers to those water molecules which are encapsulated inside the vesicles. The negative slope of this part is in accordance with a self-diffusion coefficient deriving from the Brownian motion of the vesicles. This self diffusion coefficient *D_ves_* relates to the hydrodynamic radius *r* of the vesicles according to the following equation:
(1)$$D_{ves}\;=\;\frac{kT}{6\pi \eta \;r^{}}$$

With *k* being Boltzmann’s constant, *T* the given temperature, and *η* being the viscosity constant of the aqueous surrounding. So this determination of the vesicle size is completely analogous to the the dynamic light scattering (DLS) approach, as it uses the mean square displacement of the particles (the equivalent to the self diffusion constant) in order to measure their hydrodynamic radius. However, in contrast to DLS, PFG-NMR offers the significant advantage of being focused on those structures which actually contain immobilized water, such as vesicles. Dust particles or precipitations of any kind do not have any influence on the PFG-NMR data.

On average, the self diffusion constant *D_ves_* of the given vesicles amounts to 7.1·10^−13^ m^2^/s which, assuming the viscosity of water, corresponds to a mass-averaged vesicle diameter of approximately *d* = 2*r* = 600 nm. The PFG-NMR data obtained on the samples from the aqueous phase document the presence of vesicles over the full duration of the evolution experiment. Within its given period of 160 h, the diffusion constant, as well as the vesicle diameter, shows a random variability of ±10%.

### 3.2. Peptide Formation

During the same course of regular sampling, the peptides are followed by two-dimensional liquid chromatography and mass spectrometry. A principal component analysis (Figure 4) of samples taken after 0 h and 160 h reveals the presence of a fraction of molecules which (i) is not present at the beginning of the experiment (t = 0 h); (ii) is not present in the absence of vesicles; and (iii) is not present in raw materials (solvents, eluents). Consequently, this fraction of molecules (red frame in Figure 4) only forms in the presence of vesicles and over time and therefore should contain all species which accumulated during the evolution process.

All samples taken during the experiment are carefully analyzed for oligopeptides formed by the 12 amino acids listed in Section 2.2. The analyses not only reveal the amino acid composition, but also allow for a rough estimation of the peptide concentrations. Selecting the species with the steepest concentration increase over time, a distinct group of oligopeptides, all belonging to the fraction “160 h with vesicles” (Figure 4), have been identified (Table 1).

The compositions given above do not represent the actual sequence of the amino acids. Instead, the amino acids are listed according to their polarity (most polar ones first). As a striking feature, the amino acid lysine occurs in almost all of the peptides. This may be related to its possible function as a charged head group: offering a free amino residue, it is expected to be almost quantitatively protonated in the given acidic environment (see Section 2.4) and therefore will carry at least one positive charge. Another interesting property of the longer peptides is the abundance of hydrophobic amino acids. With 60–80% occurrence, they form the largest section of the peptide chains. Most peptides also contain phenylalanine, which is the most hydrophobic amino acid within the given selection. Altogether, one can postulate that all given peptides at least have the potential to be strong amphiphiles. The comparison of the results of three subsequent evolution experiments (Table 1) shows that the shorter peptides with three and four units seem to occur repeatedly, whereas the longer peptides (five to eight units) have a tendency to show up in individual experiments only. This could mark a certain randomness in the pathway of the evolution process. Nevertheless, each individual peptide is definitely the product of a selection process as it only occurs in the presence of the vesicles and develops over time during the experiment.

Regarding its potential to integrate into vesicle membranes and to have an influence on the vesicle properties, the last species in Table 1 (Lys Ser Pro Pro Ala Ala Phe Phe) appears to be the most promising candidate. Therefore, this octapeptide was subjected to a closer analysis of its amino acid sequence using MS^n^ analysis with an iontrap-MS. Due to the low concentration of the peptide, the identification of the sequence was complicated and included some plausible considerations. According to the resulting data, the charged lysine residue occurs at the amino-terminated end of the peptide chain, most likely followed by the serine. At the carboxylic end of the chain, we most probably deal with a set of two alanine segments. That leaves two phenylalanine and two proline residues for the inner part of the chain. Due to the relatively bulky phenyl side group, it is less likely that two phenylalanine units are directly connected. Proline, on the other hand, induces a relatively stiff chain conformation and has the property to induce turns in the peptide chain. Therefore, two prolines are unlikely to be directly connected as well. That leaves an inner sequence of either Pro Phe Pro Phe or Phe Pro Phe Pro for the central part of the octapeptide. Based on these findings and conclusions, we decided on
H-Lys-Ser-Pro-Phe-Pro-Phe-Ala-Ala-OH
as the most likely structure of the given peptide. In the individual chromatogram of the peptide (Figure 5), three peaks can be observed which represent the same composition of the peptide, but obviously different sequences. All three peptides have accumulated in the evolution experiment. Therefore, we can conclude that there is a certain variability in the sequence and that the sequence given above is most likely among those accumulated species.

### 3.3. Vesicle Reassembly

Having identified an accumulated species, the main purpose of this study is to elucidate why this particular octapeptide had the potential to be selected during the evolution experiment. This question is efficiently approached by a study on the special properties of a corresponding peptide-vesicle system. Since the concentration of the octapeptide in the overall mixture is still extremely low, this task has to be performed on a reconstituted system. For this purpose, the octapeptide H-Lys-Ser-Pro-Phe-Pro-Phe-Ala-Ala-OH is commercially synthesized and added to a neat vesicle dispersion which has been prepared in the high pressure cell. The resulting peptide-vesicle system is carefully studied and compared to the original vesicle system.

When added to the original vesicles, the peptide has an immediate influence on the vesicle size (Figure 6). The volume-averaged hydrodynamic radius of the vesicles is determined from the slope of the shallow part of the Stejskal-Tanner plots. This slope corresponds to the negative diffusion coefficient associated with the Brownian motion, which in turn allows for the calculation of the vesicle size at a given temperature and solvent viscosity. From an average value of 600 nm, the diameter is suddenly reduced to 300–400 nm by the action of the peptide. Before and after the addition of the peptide, the diameter is relatively stable over time.

A special feature of the PFG-NMR measurement and of the resulting Stejskal-Tanner plot is its sensitivity for the permeation process. The variation of the level of the shallow part with the gradient pulse spacing ∆ allows for direct conclusions on the membrane permeability for the observed molecules (water in the given case). Due to the exchange-related loss of encapsulated water molecules over time, the permeation process leads to a corresponding drop of the final level of the plot with increasing pulse spacing ∆. This final level marks the fraction of water molecules which remain in the encapsulated state over the full duration of the experiment. If water molecules transfer through the membrane—either by diffusion through the intact bilayer or by flow through peptide-induced pores—this fraction is diminished and the values decrease accordingly. Studies on other vesicle systems have shown that this process follows the rules of first order kinetics. Therefore, in the given semi-logarithmic plot, it results in a linear dependence of the level of ln I/I_0_ on the pulse spacing ∆ [30,31].

In case of the neat vesicles, this drop is not observable. Instead, there is even an inverted sequence (the level for 100 ms is higher than the one for 25 ms), which results from internal water diffusion inside of the vesicle volume (Figure 7a–c). However, as soon as the peptide is added, the sequence changes drastically. Actually, the sequence now is typical for a rapid permeation process: the level for 25 ms is higher than the one for 100 ms (Figure 7d). Obviously, there is a significant loss of encapsulated water molecules in the time interval between 25 and 100 ms, which is induced by the addition of the peptide.

However, this situation changes over time. Within 1.5 h, the permeation-induced drop collapses (Figure 7d), and gradually returns to the original one observed on neat vesicles (Figure 7e,f). Finally, after 108 h storage at elevated temperature (50 °C), the original sequence is reached (Figure 8f).

The most interesting aspect of the peptide selection is the potential influence on the thermal stability of the vesicles. For the assessment of the given peptide, the PFG-NMR measurements have been repeated during a period of high-temperature storage (50 °C, 100 h), for neat vesicles as well as for the peptide-vesicle system. The result is shown in Figure 8. It reveals a significant thermal instability of the neat vesicles, as determined from the drop of the final plateau values (Figure 8, top row). The loss of the encapsulated fraction corresponds to a half-life time around 100 h. In contrast, no such loss is observed for vesicle dispersions after the addition of the peptide (Figure 8, bottom row). This demonstrates a strong stabilizing effect on the vesicles which is induced by the selected octapeptide H-Lys-Ser-Pro-Phe-Pro-Phe-Ala-Ala-OH.

## 4. Discussion

The given experiment relies on the choice of initial constituents, which of course will have a significant influence on the outcome. This includes the selection of amino acids as well as the one for the amphiphilic components. Regarding the choice of amino acids, we are aware of the fact that our restriction to use the pure L-enantiomers does not reflect the prebiotic reality. Moreover, the mixture of amphiphiles was definitely much more complex than the simple 1:1 mixture of octadecylamine and octanoic acid. However, in order to reproduce the principle of the described evolution process, these simplified conditions may well hold as a representative model for the self-evolving system.

The experimental results clearly show a significant fraction of peptides which (i) are not yet present at the beginning of the experiment and gradually form over time, and (ii) only form in the presence of the vesicles. Among these, a few peptides stand out in terms of their concentration increase over time, most prominently the octapeptide H-Lys-Ser-Pro-Phe-Pro-Phe-Ala-Ala-OH. Obviously, this peptide has been accumulated in a selection process induced by the presence of membrane vesicles. Looking at the possible mechanisms of peptide selection in detail, one can differentiate between three possible criteria favoring different types of amphiphilic peptides (Figure 9) [16]. In the following, we briefly describe the three selection criteria and discuss the possible contribution of the selected peptide H-Lys-Ser-Pro-Phe-Pro-Phe-Ala-Ala-OH:(1)Integration. Amphiphilic peptides with an amphiphilicity profile reflecting the one of the bilayer integrate into the vesicle membrane. Hereby, they gain an individual selection advantage as they are partially protected against hydrolysis and as they become less easily eluted from the vesicle zone. For this selection criterion, the effect of the peptide on the stability of the vesicle is irrelevant, hence one could call this mechanism a parasitic (or commensal) one.

The amphiphilicity profile of the selected peptide H-Lys-Ser-Pro-Phe-Pro-Phe-Ala-Ala-OH is clearly predestinated by its sequence. In the given acidic environment (pH = 3), the initial lysine group, offering an additional amino residue, will carry at least one positive charge, possibly even two since it forms the amino end of the chain. Serine is quite hydrophilic as well and can therefore contribute to the polar head of the molecule. The residual chain of the peptide is being formed by six non-polar amino acids (pairs of proline, phenyalanine, and alanine) with a clear potential to represent the hydrophobic part of the molecule. Altogether, the selected octapeptide can be regarded as strongly amphiphilic. All experimental results support its expected rapid membrane integration. The changes of the vesicle size (Figure 6) and vesicle membrane permeability (Figure 7) occur almost instantaneously and can only be interpreted by an intense and integrative peptide-membrane interaction.(2)Stabilization. In this “symbiotic” interaction, amphiphilic peptides are again being protected by vesicles, but in turn, they also stabilize the vesicle structure. This leads to a mutual advantage connected to the formation of the peptide-vesicle system. With increased stability, the vesicles could even survive several pressure cycles (“generations”), therefore giving the peptide an increased degree of protection over a longer period of time. Consequently, this “symbiotic” effect would lead to an even stronger selection advantage of the peptide.

In the given experiment, the selected peptide H-Lys-Ser-Pro-Phe-Pro-Phe-Ala-Ala-OH clearly has a protective effect on the vesicles. While the original vesicles show a significant thermal decomposition over 100 h at 50 °C (Figure 8, top row), the peptide-vesicle system seems to be stable under the same conditions (Figure 8, bottom row). Even though there is no plausible interpretation of the mechanism, the effect of thermal stabilization is obvious. An additional factor for the stability-related selection of vesicles may be the vesicle size. During the decompression step of the pressure cycling, CO_2_ bubbles are generated which have the potential to disrupt vesicles if they form inside their inner volume. Consequently, smaller vesicles have better chances of staying intact, giving peptides which reduce the vesicle size a selection advantage. This may explain the effect of the selected peptide on the vesicle diameter (Figure 6).(3)Function. The selection of peptides could go even beyond the effect of simple mechanical stabilization and lead to more complex (“functional”) survival mechanisms. A possible example may be a membrane structure which allows for the increased permeation of solvent through the membrane (like, e.g., by the formation of a channel structure [32]). In the vesicle cycle, the vesicles are generated with a natural ionic concentration gradient across the membrane which destabilizes their structure by the resulting osmotic stress. Permeation through the membrane could cause a rapid relaxation of this gradient, leading to a longer vesicle lifetime and resulting in a corresponding selection advantage for the permeation-inducing peptide.

Obviously, the selected peptide H-Lys-Ser-Pro-Phe-Pro-Phe-Ala-Ala-OH causes an increased degree of water permeation through the vesicle membranes (Figure 7), at least for a limited period of time. This period, however, is sufficient for an equilibration of concentration gradients which mainly consist of the dissolved amino acids (their overall concentration being 0.8 M in the outer aqueous phase, but nearly zero in the inner phase of the vesicles). Again, there is no clear interpretation of the mechanism and the reason why it only acts temporarily, but the effect of water permeation is significant and will probably also include dissolved components. It will lead to a rapid decrease of the osmotic pressure and therefore increase the survival rate of the vesicles.

Even though shorter peptides may have a stronger effect due to their higher concentration, the tendency of the longer peptides to be more efficient in their influence on the vesicle structure will lead to their more significant long-term accumulation.

## 5. Conclusions

The peptide H-Lys-Ser-Pro-Phe-Pro-Phe-Ala-Ala-OH selected in the described experiment may be an example of the result of a molecular evolution process occurring under very primitive and quite natural conditions. With the given sequence, it obviously integrates into the vesicle membrane, increases its thermal stability as well as its permeability, and decreases the vesicle size. All these effects can be interpreted as suitable survival strategies of the peptide-vesicle system leading to prolonged lifetimes under the given circumstances.

Even though the described mechanism of peptide selection misses the capability for identical reproduction, the mechanism of selection from a large pool of random peptides over a long period of time can be quite efficient. Both conditions are definitely fulfilled for peptides in hydrothermal sources. The entropic driving force of the process consists of the expansion and dilution of a large amount of hydrothermal products rising in the Earth’s crust, from which a small fraction is selected over a long and iterative process (Figure 10). The outcome may be a peptide-vesicle system which has developed a set of functions leading to its long-term survival. With this property, it could easily have been stable enough to travel to the Earth’s surface, either by the natural flow of tectonic fluids or by regular geyser eruptions. The further evolution process of the vesicles may have included an early stage of a primitive metabolism which could have taken advantage of concentration gradients as an initial energy source by the use of catalytically active peptide channels. Moreover, vesicles may have gained the ability to self-replicate, which opens up the possibility of following an evolutionary pathway as described by Ruiz-Mirazo and Mavelli, which follows very similar principles [33,34]. Finally, such a functional peptide-vesicle system could also have formed an ideal platform for a subsequent RNA world.

## Figures and Tables

**Figure 1 life-08-00016-f001:**
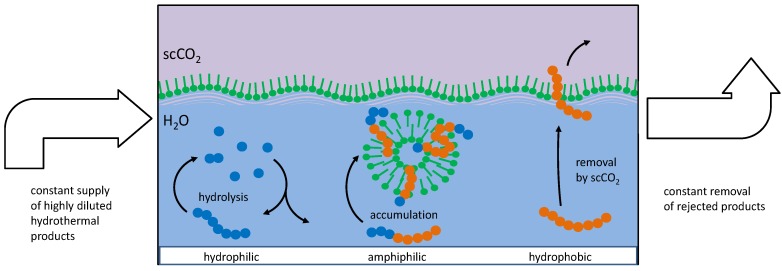
Mechanism of peptide selection and accumulation in presence of vesicles. **Left**: Peptide chains formed by hydrophilic amino acids (blue circles) will undergo little interaction with vesicles and remain in the aqueous phase where they undergo hydrolysis. **Right**: Peptide chains formed by hydrophobic amino acids (red circles) will eventually be eluted by scCO_2_. **Center**: Amphiphilic peptides will accumulate in the bilayer membrane and remain partially protected against hydrolysis and elution [16]. The thickness of the vesicle membrane is exaggerated in order to visualize the internal structure. (Figure reproduced from ref. [16]).

**Figure 2 life-08-00016-f002:**
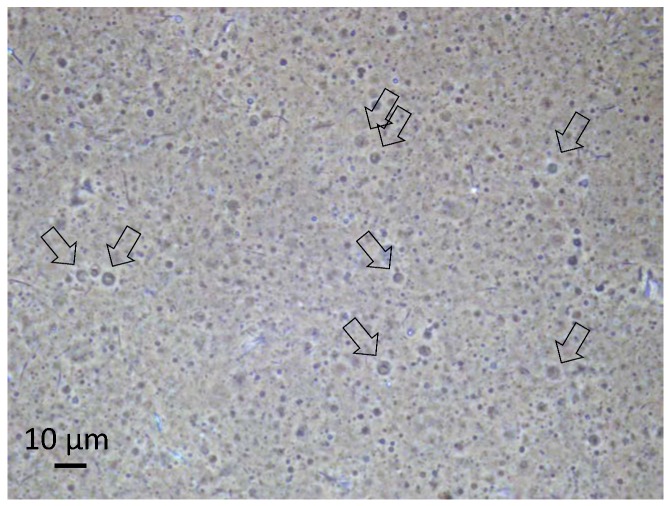
Optical micrograph of the vesicle dispersion in a sample taken during the described evolution experiment. The picture was produced under phase-contrast illumination in order to increase the visibility of the vesicles. The small fraction of vesicles larger than 2 µm can be resolved, and their membrane structure becomes apparent (arrows). Other particles in the image are primarily crystallized components. The picture was taken on a sample with a vesicle concentration of approximately 1 vol %, at neutral pH and at room temperature.

**Figure 3 life-08-00016-f003:**
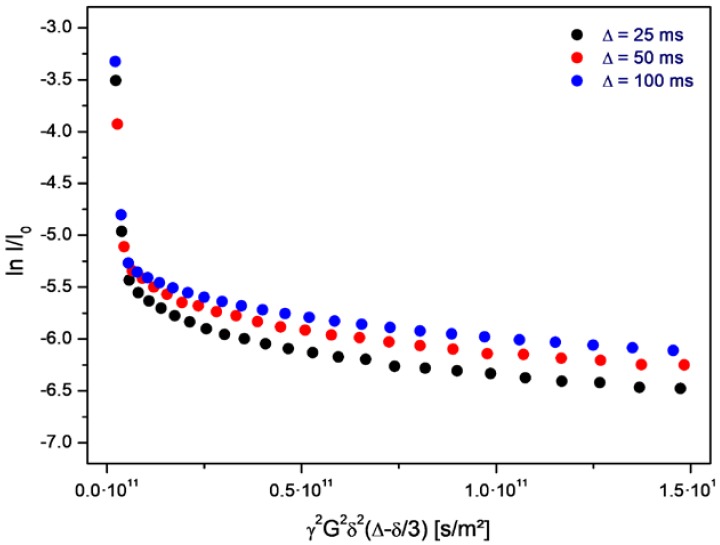
Stejskal-Tanner-plot of PFG-NMR data obtained on water molecules in a sample of the aqueous phase during an evolution experiment. The steep initial part of the echo decay corresponds to free water, and the following shallow part to encapsulated water inside the vesicles. The values for ∆ (25, 50, and 100 ms) refer to the spacing between the gradient pulses. The measurement was obtained on a sample with a vesicle concentration of approximately 0.5 vol %, at neutral pH and at room temperature.

**Figure 4 life-08-00016-f004:**
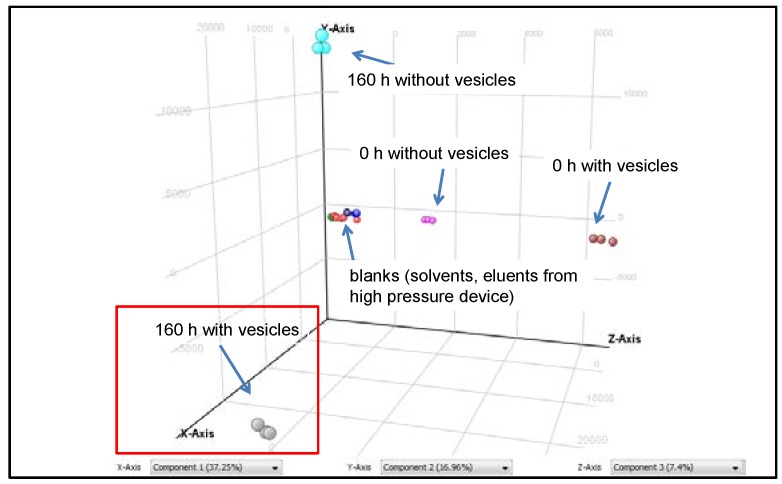
Principal component analysis of samples from an evolution experiment taken after 0 and 160 h. The red box labels the entity of molecules which are absent at t = 0 h and only form in the presence of vesicles over time.

**Figure 5 life-08-00016-f005:**
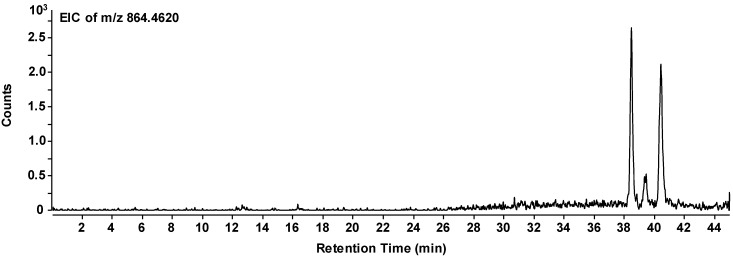
Extracted ion chromatogram of the selected mass peak representing the octapeptide composition. H-Lys-Ser-Pro-Phe-Pro-Phe-Ala-Ala-OH (*m/z* = 863.4548 g/mol for the non-protonated species). The three signals (two strong and one weak) presumably correspond to different amino acid sequences with the same overall composition.

**Figure 6 life-08-00016-f006:**
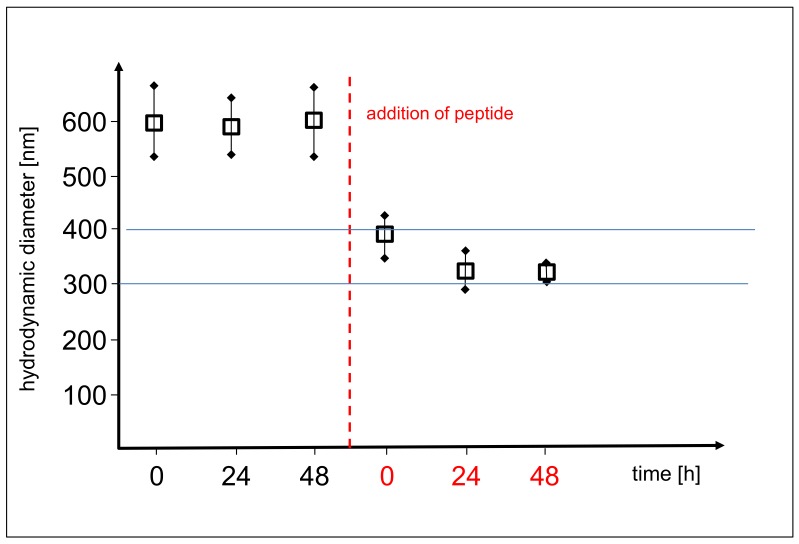
Average diameter of the vesicles before (**left**) and after the addition of the peptide H-Lys-Ser-Pro-Phe-Pro-Phe-Ala-Ala-OH (**right** of dotted line). The error bars mark the experimental variability.

**Figure 7 life-08-00016-f007:**
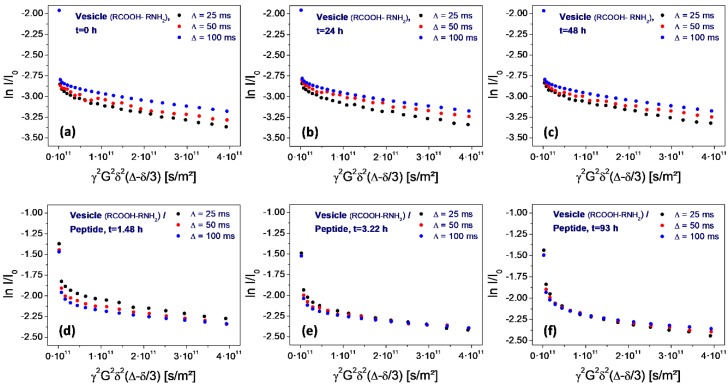
Stejskal-Tanner plots determined on water for neat vesicles (top row: **a**–**c**) and after the addition of the peptide H-Lys-Ser-Pro-Phe-Pro-Phe-Ala-Ala-OH (bottom row: **d**–**f**). As determined from the inversion of the sequence for ∆ = 25, 50, and 100 ms, the peptide causes a temporary increase of the vesicle membrane permeability for water (**d**). Within several hours, the sequence slowly returns to the original one, indicating the loss of the original permeability over time (**e**,**f**).

**Figure 8 life-08-00016-f008:**
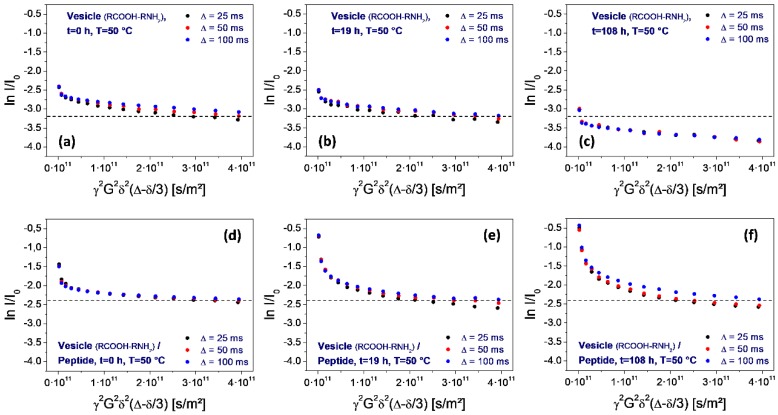
Stejskal-Tanner plots determined on water for neat vesicles (top row, **a**–**c**) and after the addition of the peptide H-Lys-Ser-Pro-Phe-Pro-Phe-Ala-Ala-OH (bottom row: **d**–**f**). Both series correspond to storage at 50 °C for 0, 19, and 100 h. As determined from the shift of the final plot level, there is a significant decomposition of the neat vesicles (top row: **a**–**c**). No such decomposition is observed on vesicles containing the peptide, revealing its stabilizing effect.

**Figure 9 life-08-00016-f009:**
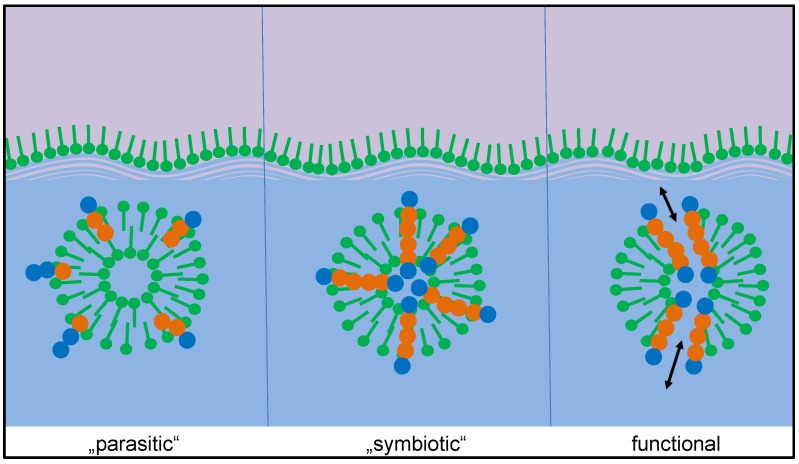
Possible targets of a selection process of peptide-vesicle systems. Integration and protection of peptide (parasitic), additional thermal stabilization of the vesicle structure (symbiotic), and introduction of a peptide-induced function (functional). The thickness of the vesicle membrane is exaggerated in order to visualize the internal structure.

**Figure 10 life-08-00016-f010:**
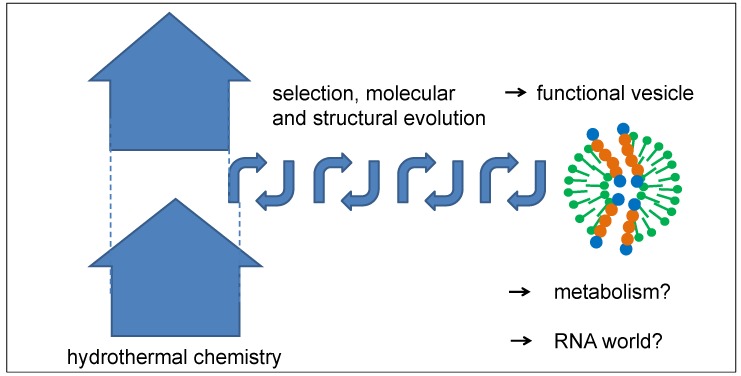
Schematic diagram of a possible structural evolution of vesicles by repeated optimization steps in tectonic fault zones. The thickness of the vesicle membrane is exaggerated in order to visualize the internal structure.

**Table 1 life-08-00016-t001:** Selected peptides showing a distinct concentration increase over time during three subsequent evolution experiments. The symbol “X” stands for a concentration high enough for an analysis of the amino acid composition. The symbol “0” means that they are below this limit.

Amino Acid Composition	Exp. 1	Exp. 2	Exp. 3
Thr Thr Pro	X	X	0
Lys Pro Pro Phe	X	X	X
Lys Lys Gly Pro Ala	X	0	X
Lys Ser Pro Ala Phe	X	0	0
Lys Pro Gly Gly Gly Phe	0	X	0
Lys Ser Pro Pro Ala Ala Phe Phe	0	0	X

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
