# Peer review of "Molecular Evolution in a Peptide-Vesicle System"

_life, 2018, doi:10.3390/life8020016_

Round 1

Reviewer 1 Report

In their manuscript the authors further develop their vesicle model, already described in previous publications ([9], [15-16], [18]), based on the hypothesis that periodic vesicle formation should have occurred in open tectonic fault systems in the early continental crust. They present the results of a new experiment performed at high temperature and pressure based on vesicles formed from two specific amphiphiles, octadecylamine and octadecanoic acid in a 1:1 mass ratio and proteinogenic amino acids in their natural L-form. The experiment yields evidence on a vesicle-induced accumulation of membrane-interacting peptide identified by liquid chromatography combined with high-resolution mass spectroscopy. In particular, the experimental results show a significant fraction of peptides which i) are not yet present at the beginning of the experiment and gradually form over time, and ii) only form in presence of the vesicles. Among these, a few peptides are sticking out in terms of their concentration increase over time, most prominently the octapeptide NH2-Lys-Ser-Pro-Phe-Pro-Phe-Ala-Ala-OH. Even though the selected shorter peptides may have a stronger effect due to their higher concentration, the selected peptide has an immediate effect on the vesicles, leading to i) reduced vesicle size, ii) increased vesicle membrane permeability, and iii) improved thermal vesicle stability.

This new article is attractive by adding new experimental data aiming to support the authors’ model.

I have the following comments and questions:

1.     Selection of the amphiphiles:

Following T Namani and DW Deamer’s view ([4]), two specific amphiphiles, octadecylamine and octadecanoic acid in a 1:1 mass ratio, were chosen as a mixture of long chain amines with long chain fatty acids accessible by Fischer-Tropsch chemistry. IA Chen and P Walde highlighted that most studies on vesicles formed from potentially prebiotic amphiphiles have so far been limited to systems containing one or two types of amphiphiles, as in the authors’ experiment. This is in contrast to the output of simulated prebiotic chemical reactions, which typically produce very heterogeneous mixtures of compounds (Chen and Walde, 2010). The authors should discuss the point.

2.     Selection of L-amino acids:

As specified line 112, only amino acids in their natural L-form were considered. The authors justify their option by saying that it was “for simplicity”. Surely, it is much simpler and easier as it allows the authors to avoid the issue of chirality. However this issue is not trivial at all. J Garay claims that any theory of origin of life that does not explain the origin of homochirality in biomolecules is not complete (Garay, 2011). The authors should mention the problem.

3.     Experimental temperature:

The authors claim that deep-reaching tectonic fault zones have favored the formation of not only amphiphilic structures leading to the formation of vesicles but also amino acids (line 61). Thus, the thermal stability of the vesicles with and without peptides was assessed by introducing storage intervals at elevated temperature between the measurements, i.e., T= 50°C (line 159). By the way, in their 2017 article [9] they specify, “tectonic fault zones offer a wide variety of reaction conditions regarding pressure, temperature, and catalytic surfaces. All reaction sites are interconnected by efficient material transport while isolated pockets allow for continuous reactions under constant conditions. Near a depth of 1 km (corresponding to T = 50 ̊C and 100 bar of hydrostatic pressure), pressure variations caused by geysers or tidal phenomena cause a cyclic formation of vesicles”. However, line 132, the authors specify that, in order to accelerate the peptide formation cycle and to induce selection pressure on the vesicles, the temperature inside the cell was kept at 120°C during the whole experiment. On the other hand, the authors specify that amino acids are expected to occur in the same kind of hydrothermal environment (line 61) first referring to WWL Marshall [22]. In his 1994 article WWL Marshal says, “Aqueous NH4HC03 solutions were reacted with C2H2, H2, and O2 (formed in situ from CaC2, Ca, and H2O2) at 200-275°C over 0.2-2 h periods to synthesize several amino acids and abundant amines. These amino acid and amine producing reactions were not observed to occur below 150°C. Amino acids and amines also were synthesized at 210°C from solutions of NH4OH, HCHO, NaCN, and H2”. They also refer to EM Andersson and NG Holm [23] who say, “The experiments were conducted at 200°C and 50 bar in Teflon coated autoclaves”. The authors should clarify this temperature issue.

4.     Reproduction:

The authors admit (line 451) that the described mechanism of peptide selection misses the capability for identical reproduction. However, they claim that the mechanism of selection from a large pool of random peptides over a long period of time can be quite efficient and that both conditions are definitely fulfilled for peptides in hydrothermal sources. It is not obvious that life spans of tectonic fault systems in the early continental crust were long enough.

5.     Evolution pathway:

In the authors’ model the selection operates on molecules such as peptides over a long and iterative process but not on lineages of vesicles. Thus, Darwinian evolution is not operating as such. When the authors claim that the outcome may be a peptide-vesicle system, which has developed a set of functions leading to its long-term survival it does not mean that a population of selected stable enough vesicles was capable for identical reproduction and inheritance of its characteristics. Thus, the evolution pathway to vesicles with “a primitive metabolism which could have taken advantage of concentration gradients as an initial energy source by the use of catalytically active peptide channels” is not straightforward as the authors describe it all. The authors’ claim should be less affirmative.

References:

Chen IA, Walde P. From self-assembled vesicles to protocells. Cold Spring Harbor Perspectives in Biology. 2010;2:a002170.

Garay J. Active centrum hypothesis: The origin of chiral homogeneity and the RNA-world. BioSystems. 2011;103:1–12.

Author Response

Dear Academic Editor, dear Reviewers,

Thank you for again considering our manuscript for publication in life. As desired, we have now revised our present manuscript according to the comments and recommendations of the review reports. In the following, we want to respond to the comments and summarize our changes and additions to the text.

Reviewer 1

Comments and Suggestions for Authors

In their manuscript the authors further develop their vesicle model, already described in previous publications ([9], [15-16], [18]), based on the hypothesis that periodic vesicle formation should have occurred in open tectonic fault systems in the early continental crust. They present the results of a new experiment performed at high temperature and pressure based on vesicles formed from two specific amphiphiles, octadecylamine and octadecanoic acid in a 1:1 mass ratio and proteinogenic amino acids in their natural L-form. The experiment yields evidence on a vesicle-induced accumulation of membrane-interacting peptide identified by liquid chromatography combined with high-resolution mass spectroscopy. In particular, the experimental results show a significant fraction of peptides which i) are not yet present at the beginning of the experiment and gradually form over time, and ii) only form in presence of the vesicles. Among these, a few peptides are sticking out in terms of their concentration increase over time, most prominently the octapeptide NH2-Lys-Ser-Pro-Phe-Pro-Phe-Ala-Ala-OH. Even though the selected shorter peptides may have a stronger effect due to their higher concentration, the selected peptide has an immediate effect on the vesicles, leading to i) reduced vesicle size, ii) increased vesicle membrane permeability, and iii) improved thermal vesicle stability.

This new article is attractive by adding new experimental data aiming to support the authors’ model.

I have the following comments and questions:

1.     Selection of the amphiphiles:

Following T Namani and DW Deamer’s view ([4]), two specific amphiphiles, octadecylamine and octadecanoic acid in a 1:1 mass ratio, were chosen as a mixture of long chain amines with long chain fatty acids accessible by Fischer-Tropsch chemistry. IA Chen and P Walde highlighted that most studies on vesicles formed from potentially prebiotic amphiphiles have so far been limited to systems containing one or two types of amphiphiles, as in the authors’ experiment. This is in contrast to the output of simulated prebiotic chemical reactions, which typically produce very heterogeneous mixtures of compounds (Chen and Walde, 2010). The authors should discuss the point.

This is a good point, and we agree that this issue should be included in our discussion. We definitely want to make clear that we do not believe that peptide evolution actually started with enantiomerically pure amino acids. Instead, we regard our experiment as a suitable model for the actual process. Therefore, we have added a corresponding statement to the initial part of the Discussion section:

“The given experiment relies on the choice of initial constituents which of course will take a significant influence on the outcome. This includes the selection of amino acids as well as the one for the amphiphilic components. Regarding the choice of amino acids, we are aware of the fact that our restriction to use the pure L-enantiomers does not reflect the prebiotic reality. (…) However, in order to reproduce the principle of the described evolution process, these simplified conditions may well hold as a representative model for the self-evolving system.”

2.     Selection of L-amino acids:

As specified line 112, only amino acids in their natural L-form were considered. The authors justify their option by saying that it was “for simplicity”. Surely, it is much simpler and easier as it allows the authors to avoid the issue of chirality. However this issue is not trivial at all. J Garay claims that any theory of origin of life that does not explain the origin of homochirality in biomolecules is not complete (Garay, 2011). The authors should mention the problem.

Of course the intention of our experiment is not to deliver a complete explanation for the origin of life. Instead, it is meant as a model for the type of molecular and structural evolution process which may have led to functional vesicles. This, however, can in fact be done based on a simplified version of the actual mixture of prebiotic compounds. In order to make this point clearer, we have added the statement (see also above):

“The given experiment relies on the choice of initial constituents which of course will take a significant influence on the outcome. This includes the selection of amino acids as well as the one for the amphiphilic components. (…) Moreover, the mixture of amphiphiles was definitely much more complex than the simple 1:1 mixture of octadecylamine and octanoic acid. However, in order to reproduce the principle of the described evolution process, these simplified conditions may well hold as a representative model for the self-evolving system.”

3.     Experimental temperature:

The authors claim that deep-reaching tectonic fault zones have favored the formation of not only amphiphilic structures leading to the formation of vesicles but also amino acids (line 61). Thus, the thermal stability of the vesicles with and without peptides was assessed by introducing storage intervals at elevated temperature between the measurements, i.e., T= 50°C (line 159). By the way, in their 2017 article [9] they specify, “tectonic fault zones offer a wide variety of reaction conditions regarding pressure, temperature, and catalytic surfaces. All reaction sites are interconnected by efficient material transport while isolated pockets allow for continuous reactions under constant conditions. Near a depth of 1 km (corresponding to T = 50 ̊C and 100 bar of hydrostatic pressure), pressure variations caused by geysers or tidal phenomena cause a cyclic formation of vesicles”. However, line 132, the authors specify that, in order to accelerate the peptide formation cycle and to induce selection pressure on the vesicles, the temperature inside the cell was kept at 120°C during the whole experiment. On the other hand, the authors specify that amino acids are expected to occur in the same kind of hydrothermal environment (line 61) first referring to WWL Marshall [22]. In his 1994 article WWL Marshal says, “Aqueous NH4HC03 solutions were reacted with C2H2, H2, and O2 (formed in situ from CaC2, Ca, and H2O2) at 200-275°C over 0.2-2 h periods to synthesize several amino acids and abundant amines. These amino acid and amine producing reactions were not observed to occur below 150°C. Amino acids and amines also were synthesized at 210°C from solutions of NH4OH, HCHO, NaCN, and H2”. They also refer to EM Andersson and NG Holm [23] who say, “The experiments were conducted at 200°C and 50 bar in Teflon coated autoclaves”. The authors should clarify this temperature issue.

In fact, we see the versatility of the proposed reaction environment as a big advantage for the diversity of chemical reactions. That includes the temperature and pressure conditions, which both can vary over a very wide range. We believe that the formation of amino acids, which requires very high temperatures and pressures (as proposed by Marshall in 1994), occurs at a much lower depth, e.g. at – 5 km in the Earth’s crust. The resulting amino acids could then travel to higher levels by simple transportation processes, such as flow of water and/or carbon dioxide. To clarify this aspect, we have added a statement to the introductory section, reading:

“In the same hydrothermal environment, amino acids are expected to occur [22-24], which are being formed at higher pressure and temperature conditions and are being spread by convectional flow of the fluid media.”

Regarding the choice of 50°C for the assessment of the vesicle stability, this is just a random choice of a testing condition in order to determine relative thermal stability differences between different vesicle varieties.

4.     Reproduction:

The authors admit (line 451) that the described mechanism of peptide selection misses the capability for identical reproduction. However, they claim that the mechanism of selection from a large pool of random peptides over a long period of time can be quite efficient and that both conditions are definitely fulfilled for peptides in hydrothermal sources. It is not obvious that life spans of tectonic fault systems in the early continental crust were long enough.

The lack of identical reproducibility on one side and the efficiency of a selection on the other do not necessarily contradict each other. There may be many alternative peptides with differing amino acid sequences which fulfill a given purpose (e.g. a stabilizing function) to the same degree. As we know from Biology, evolution can proceed via very different pathways, but never­theless may lead to final optimization.

Regarding the lifetime of tectonic fault zone structures, recent examples (as the “Bayrischer Pfahl” in Bavaria, Germany) are known to have been stable for more than 100 million years. Other known examples may only have existed for 10 million years. On early Earth, lifetimes may have been somewhat shorter, but still would have given plenty of room for evolution processes.

5.     Evolution pathway:

In the authors’ model the selection operates on molecules such as peptides over a long and iterative process but not on lineages of vesicles. Thus, Darwinian evolution is not operating as such. When the authors claim that the outcome may be a peptide-vesicle system, which has developed a set of functions leading to its long-term survival it does not mean that a population of selected stable enough vesicles was capable for identical reproduction and inheritance of its characteristics. Thus, the evolution pathway to vesicles with “a primitive metabolism which could have taken advantage of concentration gradients as an initial energy source by the use of catalytically active peptide channels” is not straightforward as the authors describe it all. The authors’ claim should be less affirmative.

We fully agree that, in fact, there is no identical reproduction of the vesicles. However, in some way the information about the structure is actually inherited. The reason is the following: the most stable vesicles remain in the vesicle-forming layer for an extended period of time. Unstable vesicles on the other hand disintegrate. Following this disintegration, their chemical constituents are either lost to the carbon dioxide phase or very likely being hydrolyzed. The components of the stable vesicles, on the other hand, are preserved and remain in the vesicle forming layer. On the long run, all components forming stable vesicles are being accumulated in the vesicle forming zone. So actually there is some sort of inheritance for the characteristics of successful vesicles. Therefore, we would like to keep this claim.

All changes in the manuscript can be followed by the tracking mode of MS Word. We hope that, with the additions and changes made, our manuscript is now acceptable for publication in LIFE.

We are looking forward to further comments and recommendations.

With kind regards                                                                          Christian Mayer

Reviewer 2 Report

It was a pleasure reading the manuscript. It is well written and a valuable contribution to the field of protocell research. The results obtained are carefully discussed. The PEG-NMR method used can be applied for other similar systems as well. It is one of the few current attempts to link the potentially prebiotic peptide formation with the potentially prebiotic assembly of amphiphiles into membranous aggregates. Therefore, I recommend acceptance of the manuscript after considering a few minor points.

1.       The general concept of the work is similar to the scenarios outlined by Ruiz-Mirazo and Mavelli:

On the way towards 'basic autonomous agents': Stochastic simulations of minimal lipid-peptide cells By: Ruiz-Mirazo, Kepa; Mavelli, Fabio, BIOSYSTEMS   Volume: 91   Issue: 2   Pages: 374-387   Published: FEB 2008

2.       The drawing of the vesicles in the figures of the manuscript is not optimal and may be misleading. Usually the internal aqueous diameter is much larger if compered to the vesicle bilayer thickness. See point 13 below.

3.       Figure 1: Is this drawing identical with the one of ref. [16]?  Please clarify!

4.       Line 97: … allow for the formation of vesicles ..

5.       Line 99 and ref.[29]: Namani

6.       Line 110: … hydrophobic ones …

7.       Line 124: Why 0.067 M? Why not a different concentration?

8.       General: Abbreviation used for the peptides: A dipeptide with free amino group and free carboxylic acid usually is abbreviated as H-X-Y-OH, e.g. H-Lys-Ser-….-Ala-OH. This indicated that the amino group is free (not protected) and the carboxylic acid is also free (not protected). The way the peptides are abbreviated is not ideal. If kept like it is given in the manuscript, I suggest to specifying this abbreviation.

9.       Figure 2: I would suggest to adding a comment on the other aggregates present, those which are different from vesicles. I would also add the concentration and pH in the figure legend. Obviously the visualization of the vesicles was made at room temperature.

10.   Figure 3: I suggest adding the concentration, pH and T to the legend.

11.   Table 1: I would separate the different amino acids by commas

12.   Figure 5. How does the commercial octapeptide with known sequence compare to the ones obtained in the reaction?

Figure 10: This figure is conceptually important. It could be improved by drawing a more realistic sketch of an unilamellar vesicle (see for example Figure 2 in Life 2015, 5, 1239-1263; doi:10.3390/life5021239) and by reducing the size of the arrows on the left hand side. Or does the large size have a special meaning?

Author Response

Editor-in-Chief
LIFE                                                                                                              May 17th  2018

Dear Academic Editor, dear Reviewers,

Thank you for again considering our manuscript for publication in life. As desired, we have now revised our present manuscript according to the comments and recommendations of the review reports. In the following, we want to respond to the comments and summarize our changes and additions to the text.

Reviewer 2

It was a pleasure reading the manuscript. It is well written and a valuable contribution to the field of protocell research. The results obtained are carefully discussed. The PEG-NMR method used can be applied for other similar systems as well. It is one of the few current attempts to link the potentially prebiotic peptide formation with the potentially prebiotic assembly of amphiphiles into membranous aggregates. Therefore, I recommend acceptance of the manuscript after considering a few minor points.

1.       The general concept of the work is similar to the scenarios outlined by Ruiz-Mirazo and Mavelli:

On the way towards 'basic autonomous agents': Stochastic simulations of minimal lipid-peptide cells By: Ruiz-Mirazo, Kepa; Mavelli, Fabio, BIOSYSTEMS   Volume: 91   Issue: 2   Pages: 374-387   Published: FEB 2008

This is actually a very good point, thank you for making us aware of this article. Actually, the model which is proposed by Ruiz-Mirazo and Mavelli is based on minimal autonomous and self-replicating lipid-peptide cells could represent a fascinating second step of evolution, following approximately the same principles. The vesicles in our model are neither autonomous nor self-replicating, but the selection process proposed in our manuscript could in fact lead to this property. Therefore, we now have mentioned this work together with two quotations right at the end of the manuscript:

“Moreover, vesicles may have gained the ability to self-replicate, which opens up the possibility to follow an evolutionary pathway as described by Ruiz-Mirazo and Mavelli which follows very similar principles.”

The new references

1       Mavelli, F., Ruiz-Mirazo, K. Stochastic simulations of minimal self-reproducing cellular systems. Phil Trans. R. Soc. London B. 2007, 362, 1789-1802.

2       Ruiz-Mirazo, K. Mavelli, F. On the way towards “basic autonomous agents”: stochastic simulations of minimal lipid-peptide cells. BioSystems 2008, 91, 374-387.

were added as [33] and [34].

2.       The drawing of the vesicles in the figures of the manuscript is not optimal and may be misleading. Usually the internal aqueous diameter is much larger if compered to the vesicle bilayer thickness. See point 13 below.

That is right. However, being drawn in the correct relation, the vesicle diameter would have to be one hundred times larger than the membrane thickness. This given, all details on the membrane structure would be lost. Therefore, we would like to stick to the drawings as they are, but add a comment to the legend stating that the thickness of the vesicle membranes are exaggerated. The new figure captions contain this statement.

3.       Figure 1: Is this drawing identical with the one of ref. [16]?  Please clarify!

Yes, this drawing is identical to the one in Ref. [16]. It represents the key idea for peptide selection, therefore is crucial for the contents of both papers. But we agree that this should be mentioned. We have added a corresponding statement to the figure caption.

4.       Line 97: … allow for the formation of vesicles ..

Thanks for pointing that out, we have corrected the text accordingly.

5.       Line 99 and ref.[29]: Namani

Again, our mistake. Thanks for pointing that out, we have corrected the text accordingly.

6.       Line 110: … hydrophobic ones …

Thanks for pointing that out, we have corrected the text accordingly.

7.       Line 124: Why 0.067 M? Why not a different concentration?

With the given concentrations, we were touching the solubility limit for some amino acids. So the given value is the result of maximizing the concentration.

8.       General: Abbreviation used for the peptides: A dipeptide with free amino group and free carboxylic acid usually is abbreviated as H-X-Y-OH, e.g. H-Lys-Ser-….-Ala-OH. This indicated that the amino group is free (not protected) and the carboxylic acid is also free (not protected). The way the peptides are abbreviated is not ideal. If kept like it is given in the manuscript, I suggest to specifying this abbreviation.

Okay, accepted. We have changed the abbreviation accordingly.

9.       Figure 2: I would suggest to adding a comment on the other aggregates present, those which are different from vesicles. I would also add the concentration and pH in the figure legend. Obviously the visualization of the vesicles was made at room temperature.

We have added the comment and all parameters to the figure caption.

10.   Figure 3: I suggest adding the concentration, pH and T to the legend.

We have added all parameters to the figure caption.

11.   Table 1: I would separate the different amino acids by commas

Sorry, here we find commas distracting. If possible, we would like to leave the table as it is.

12.   Figure 5. How does the commercial octapeptide with known sequence compare to the ones obtained in the reaction?

We are very sure that the octapeptide we have had synthesized commercially is one of the three varieties which are shown in Fig. 4. Therefore, we positively believe that it has been one of the selected peptide specimen.

Figure 10: This figure is conceptually important. It could be improved by drawing a more realistic sketch of an unilamellar vesicle (see for example Figure 2 in Life 2015, 5, 1239-1263; doi:10.3390/life5021239) and by reducing the size of the arrows on the left hand side. Or does the large size have a special meaning?

Again, we would like to stick tot he actual version, but now we have stated that stated that the thickness of the membrane is exaggerated in the figure caption.

The large size of the arrows is intentional. It is meant to indicate that the overall volume of material carrying hydrothermal products is very large in comparison to the selected fraction.

All changes in the manuscript can be followed by the tracking mode of MS Word. We hope that, with the additions and changes made, our manuscript is now acceptable for publication in LIFE.

We are looking forward to further comments and recommendations.

With kind regards                                                                          Christian Mayer

Round 2

Reviewer 1 Report

I thank the authors for their responses to my comments and the changes and additions to the text.

The authors agree that, in fact, there is no identical reproduction of the vesicles. However, they add that, in some way the information about the structure is actually inherited. The reason is that the components of the stable vesicles are preserved and remain in the vesicle forming layer and thus, on the long run, all components forming stable vesicles are being accumulated in the vesicle forming zone.

I don't see how separate lineages of vesicle populations could emerge without the reproduction of the vesicles. In the absence of separate lineages natural selection cannot operate. Thus, I still maintain that further evolution is not straightforward within the model.